# A Novel Mathematical Model That Predicts the Protection Time of SARS-CoV-2 Antibodies

**DOI:** 10.3390/v15020586

**Published:** 2023-02-20

**Authors:** Zhaobin Xu, Dongqing Wei, Hongmei Zhang, Jacques Demongeot

**Affiliations:** 1Department of Life Science, Dezhou University, Dezhou 253023, China; 2School of Life Sciences and Biotechnology, Shanghai Jiao Tong University, Shanghai 200030, China; 3Laboratory AGEIS EA 7407, Team Tools for e-Gnosis Medical, Faculty of Medicine, University Grenoble Alpes (UGA), 38700 La Tronche, France

**Keywords:** SARS-CoV-2, antibody dynamics, vaccine, protection time, mathematical modeling

## Abstract

Infectious diseases such as SARS-CoV-2 pose a considerable threat to public health. Constructing a reliable mathematical model helps us quantitatively explain the kinetic characteristics of antibody-virus interactions. A novel and robust model is developed to integrate antibody dynamics with virus dynamics based on a comprehensive understanding of immunology principles. This model explicitly formulizes the pernicious effect of the antibody, together with a positive feedback stimulation of the virus–antibody complex on the antibody regeneration. Besides providing quantitative insights into antibody and virus dynamics, it demonstrates good adaptivity in recapturing the virus-antibody interaction. It is proposed that the environmental antigenic substances help maintain the memory cell level and the corresponding neutralizing antibodies secreted by those memory cells. A broader application is also visualized in predicting the antibody protection time caused by a natural infection. Suitable binding antibodies and the presence of massive environmental antigenic substances would prolong the protection time against breakthrough infection. The model also displays excellent fitness and provides good explanations for antibody selection, antibody interference, and self-reinfection. It helps elucidate how our immune system efficiently develops neutralizing antibodies with good binding kinetics. It provides a reasonable explanation for the lower SARS-CoV-2 mortality in the population that was vaccinated with other vaccines. It is inferred that the best strategy for prolonging the vaccine protection time is not repeated inoculation but a directed induction of fast-binding antibodies. Eventually, this model will inform the future construction of an optimal mathematical model and help us fight against those infectious diseases.

## 1. Introduction

The COVID-19 epidemic has caused more than 630 million infections and over 6.5 million deaths worldwide by the end of 2022, and it can hardly disappear in a short time based on observation [1,2,3]. Until now, vaccination has been the only approach to fighting SARS-CoV-2 infections. As the global promotion of the SARS-CoV-2 vaccine continues, we are gaining a better understanding of the antibodies triggered by vaccines and natural infections. It is gradually becoming apparent that antibodies against SARS-CoV-2, whether acquired from natural infection or triggered by vaccination, decay over time, with a concomitant decrease in protective efficiency. Unlike antibodies from vaccinations such as smallpox [4], antibodies against SARS-CoV-2 do not provide durable protection [5].

A significant portion of the population is reluctant to be vaccinated, according to a public poll survey [6,7]. The first concern comes from the side effects of vaccination, such as blood clots [8,9]. The second concern is this deficiency of the long-lasting protection effect brought by vaccination since many clinical reports indicate that the vaccine’s protection effect is declining over time [10,11,12,13]. Based on the clinical data, a significant decrease in the immune effect of the vaccine against reinfection has been noticed. The SARS-CoV-2 vaccine still shows an overall positive effect on protection currently. However, more people are concerned about how long the protection can be provided by the antibodies triggered by their immune response [5,14]. There is an urgent need to evaluate the protection time of neutralizing antibodies quantitatively.

Constructing a reliable mathematical model helps us quantitatively explain the virus dynamics in the host body, which could provide a reasonable prediction toward those sensitive concerns faced by the public. Researchers have conducted many studies on virus infections using computational approaches [15,16].

Specifically, depending on whether the immune response is integrated into the models, traditional cellular models can be divided into two categories. The first set contains models that do not include the immune response [17,18,19,20,21], that is, traditional viral kinetic models, whose following equation can be classically expressed as:(1)$$\frac{dT}{dt}=sT-dT-\beta TV$$
(2)$$\frac{dI}{dt}=\beta TV-\delta I$$
(3)$$\frac{dV}{dt}=pI-cV$$
where *T* denotes the target cells or the number of susceptible cells; *I* denotes the number of cells where the infection has occurred; and *V* denotes the number of viruses. This simplified model is wildly used to study the virus dynamics upon infection. While admitting its advantages in recapturing the viral dynamics in virus infection, concerns about this model are proposed. The limitation of this model originates from the absence of a connection to the body’s immune response. The main driving force behind eliminating the virus is activating the adaptive immune response by lymphocyte B and T cells, not the depletion of the susceptible cells. Therefore, the estimation based on those depletion mechanisms is skeptical, as it would return a much smaller initial susceptible cell number. The number of susceptible cells should be higher than the number estimated by several orders. Mandating the estimation would inevitably lead to underestimating *T_0_*, where *T_0_* represents the number of initial susceptible cells. The *T_0_* term is similar to the initial number of susceptible people when the *SIR* (susceptible, infection, recovery) model is used to predict the epidemic trend [22,23]. The estimated *T_0_* values vary greatly due to the differences in the fitting data, which significantly reduces their reliability.

The second set consists of models that consider the immune response, where the classical expression of the model considering antibody binding is displayed as follows [24,25,26,27,28,29,30]:(4)$$\frac{dT}{dt}=\chi D-\beta TV$$
(5)$$\frac{dI_{1}}{dt}=\beta TV-kI_{1}$$
(6)$$\frac{dF}{dt}=\omega V-\alpha F$$
(7)$$\frac{dD}{dt}=\delta I_{2}-\chi D$$
(8)$$\frac{dI_{2}}{dt}=kI_{1}-\delta I_{2}$$
(9)$$\frac{dA}{dt}=fV-hA\;$$
(10)$$\frac{dV}{dt}=\frac{p}{1+\varepsilon_{1}F\;}I_{2}-cV-\gamma \;TV-\kappa AV$$
where *T* denotes the overall number of susceptible cells; *A* denotes the number of antibodies; *V* denotes the amount of virus; *D* denotes the number of dead cells; *F* denotes the amount of interferon in nonspecific immunity; $I_{1}$ denotes the number of cells infected in class 1 (cells infected without viral multiplication); and $I_{2}$ denotes the number of cells infected in class 2 (cells infected with viral multiplication). This model effectively combines antibody dynamics with viral changes. It performs well in recapturing and explaining the dynamic interaction between the virus and the host immune system. However, the kinetic properties of antibodies have not been explicitly integrated into this model. Different antigen-binding antibodies will exhibit different kinetics, and this diversity in binding behavior is an essential factor influencing the host-pathogen interaction. The difference in the antibody binding capacity would lead to a significant variation in the virus dynamics. However, this classical model does not explicitly formulize the antibody term.

Although mathematical modeling of antibody kinetics has been significantly advanced and developed in recent years, there is still great potential to be further improved. Therefore, a new mathematical model of antibody dynamics is proposed by us. This model displays good fitness in explaining some puzzling phenomena, especially in the case of SARS-CoV-2 infection, with improvements to previous models in several aspects.

First: New immunology responses are formulized and integrated into this model based on biochemistry principles. We explicitly formulize the virtual effect of antibodies on eliminating pathogenic microorganisms while noticing the stimulation effect of pathogenic antigens on antibodies [31]. This positive feedback stimulation is explicitly represented in a mathematical function as well.

Second: The dynamic model developed hypothesizes that environmental factors maintain memory cell levels. The natural attenuation of antibody levels can be recaptured in this model. Furthermore, their discontinuous decay trend can also be captured with the introduction of environmental antigens. Antigen-like substances in the environment can maintain a specific concentration of B cells or T cells at a certain level. This can explain why some vaccines can provide lifelong protection.

Third: The antigen–antibody binding process is represented as reversible in our model. Instead of using the equilibrium constant, the binding dynamic is described in both the binding and reverse dissociation reactions.

Based on the basic principles of immunology, we established the theoretical hypothesis of antibody kinetics. The results and methods sections illustrate specific principles and rationales in detail. In the end, this antibody kinetic model proposes some possible mechanisms underlying some real-world scenarios:How are memory cells maintained?How does our immune system screen for antibodies with a strong binding affinity?Why do people who get influenza and other vaccines have a lower mortality rate from SARS-CoV-2?How can we effectively calculate the duration of protection of a specific antibody?Why are some recovered patients retested as positive cases without infections from other people?Why do vaccinations show considerable differences in protection efficiency?How can we improve the protective efficiency and duration of vaccines?

## 2. Materials and Methods

### 2.1. Mathematical Representation of the Antibody Production Process

A simple mathematical representation of the immune response is described in the diagram in Figure 1 below.

Here, *x* denotes the number of antibody–antigen (virus) complexes, *y* denotes the total number of antibodies, and *z* denotes the number of viruses. Six processes are displayed in our model. The first reaction represents the virus’s proliferation or replication with a reaction constant $k_{1}$. The second reaction represents the binding reaction between virus and antibody, with a forward reaction constant $k_{2}$ and reverse constant $k_{-2}$. The third reaction represents removing the antibody–virus complex with a reaction constant $k_{3}$ with the help of natural killer (NK) cells [32]. The fourth reaction represents the induction of a new antibody by the antibody–virus complex with a kinetic constant $k_{4}$. In immunology, those virus–antibody complexes are on the surface of B-cells since the antibodies are initially produced by B-cells and will attach to the plasma membrane of B-cells. Those complexes would further bind to the helper cells because the antibody has another structure binding region toward those receptors. Those helper cells will present the antigen part, a virus in this case, to the T-cells. The physical placement should be such that B-cells bind to those helper cells and further present themselves close to T-cells. The T-cells will handle those antigen substances; if those substances are not self-originated, they will secrete signal molecules to promote the proliferation or division of B-cells that attach to them. Therefore, B-cells finally proliferate, along with the antibodies their B-cells generate. The fifth reaction represents the degradation of the virus with a constant $k_{5}$. The sixth reaction represents the degradation of antibodies with a rate constant $k_{6}$.

The differential equations based on those reactions are derived as below:

We established the following equations to represent the proliferation process of antibodies:(11)$$\frac{dx}{dt}=k_{2}yz-k_{-2}x-k_{3}x,$$
(12)$$\frac{dy}{dt}=k_{-2}x-k_{2}yz+k_{4}x-k_{6}y,$$
(13)$$\frac{dz}{dt}=k_{-2}x-k_{2}yz-k_{5}z+k_{1}z.$$

### 2.2. Mathematical Modeling including Environmental Antigens

The equations introduced in Section 2.1 describe antibody dynamics in the presence of the associated virus. In reality, other environmental antigens can induce antibody coupling and production. To further account for this possibility, we add a new set of equations as shown in Figure 2 below:

Where *p* denotes antigen-like substances in the environment that are supposed to be constant; *q* denotes antibodies bound to antigen-like substances in the environment. Reaction 7 represents the binding reaction between antibodies and environmental antigenic substances with a forward constant $k_{7}\;$ and a reverse constant $k_{-7}$. Reaction 8 represents the removal of the antibody–antigen complex *q* with a reaction rate $k_{3}$. Reaction 9 represents the induction of a new antibody by *q*. Therefore, a new set of equations is derived as follows:(14)$$\frac{dx}{dt}=k_{2}yz-k_{-2}x-k_{3}x,$$
(15)$$\frac{dy}{dt}=k_{-2}x-k_{2}yz+k_{4}x-k_{6}y-k_{7}py+k_{-7}q+k_{4}q,$$
(16)$$\frac{dz}{dt}=k_{-2}x-k_{2}yz-k_{5}z+k_{1}z,$$
(17)$$\frac{dp}{dt}=0,$$
(18)$$\frac{dq}{dt}=k_{7}py-k_{-7}q-k_{3}q.$$

The antibody level will eventually drop to zero based on the first model described in Equations (11)–(13). Since the decaying coefficient $k_{6}$ is more significant than zero, the antibody will finally fade away quickly. However, as we all know, some antibodies can persist in the human body for a long time and provide lifelong protection. It forms the basis for a vaccine. Therefore, “environmental antigen-like substances” are introduced into the second model. They could be food-resourced, air-resourced, or even self-resourced. The presentation of those substances to T-cells would give weak signals to proliferate the memory B-cells or T-cells. The detailed physical background is introduced in Section 3.1.

### 2.3. A Simplified Model Simulating the Proliferation of Antibodies with Different Binding Kinetic Characteristics by the Immune System


(19)
$$\frac{dx_{i}}{dt}=Ki_{\mathrm{on}}y_{i}z-Ki_{\mathrm{off}}x_{i}-k_{3}x_{i},$$



(20)
$$\frac{dy_{i}}{dt}=Ki_{off}x_{i}-Ki_{\mathrm{on}}y_{i}z+k_{4}x_{i}-k_{6}\left(y_{i}-w_{i}\right),$$



(21)
$$\frac{dz}{dt}=\sum_{i=1}^{n}(Ki_{\mathrm{off}}x_{i}-Ki_{\mathrm{on}}y_{i}z)-k_{5}z+k_{1}z.$$


Here, $x_{i}$ denotes the amount of the *i*-th antibody that binds to the viral antigen, $y_{i}$ denotes the total number of the *i*-th antibody, $w_{i}$ is a constant that indicates the threshold for maintaining the *i*-th antibody by the antigen-like substances in the environment. $Ki_{\mathrm{on}}$ denotes the binding constant of the free antibody *i* and the viral antigen. $Ki_{\mathrm{off}}\;$ denotes the reaction coefficient for the dissociation reaction of the antibody *i* viral complex. After we finish the second model, we want to show how our immune system selects good-binding antibodies. Therefore, we introduce the third model. The third model is a combination of the first model and second model. The difference between the third and second models is that they do not explicitly integrate environmental factors. Since each kind of antibody will have a corresponding environmental counterpart, it would be too complicated if we explicitly integrated all of them. Therefore, we use a simplified term $w_{i}$ to represent the minimal antibody level threshold maintained by its corresponding environmental-antigenic substances. Each antibody has different binding kinetics toward the virus. The modeling result demonstrates the dynamic process by which our immune system selects those fast-binding antibodies over slower ones.

The above three models are numerically solved by MATLAB using the ode15s function [33].

## 3. Results

### 3.1. Physical Mechanism behind This Approach and the Underlying Relationships among the Three Models

Our antibody model is based on the following four rationales.

The first is the stimulus of the antibody–antigen complex on the regeneration of its corresponding antibody. The T-cells will induce the proliferation of the B-cells through the antigen-presenting process. Eventually, the corresponding antibody can be reproduced following the magnification of antibody-generating B-cells [34]. Although this regulation is critical in initializing the adaptive immune response, it was not explicitly represented in previous studies [24,25,26,27,28,29,30]. The $k_{4}x$ term in Equation (12) expresses this positive feedback regulation. Integrating feedback regulation into the classical representation of biological reactions allows us to simulate reactions more comprehensively. Specifically, when we refer to the concentration alternation of antibody and virus, the change of antibody in Equation (12) should be $\;k_{-2}x-k_{2}yz-k_{6}y$, if we use chemical reactions to describe it. However, in this way, we ignore the virus–antibody complex’s stimulation of the regeneration of antibodies. In immunology, those virus–antibody complexes will proliferate the corresponding antibody with the help of T-cells after an antigen presentation [35]. Therefore, we use an external $k_{4}x$ term to model this positive feedback effect.

The second is the kinetic relationship between antibodies and antigens. This antigen–antibody binding is represented in the second process in the first diagram. The model is endowed with more dynamic characteristics with the introduction of a reversible reaction.

The third is the pernicious effect of antibodies on antigens, which is expressed as the $-k_{3}x$ term in Equation (11). It has been recognized that the antibody–antigen complex can be efficiently removed with the help of functional immune cells such as NK (natural killer) cells [32].

The fourth is the introduction of environmental antigenic substances. We hypothesize that the environmental antigen contributes to the maintenance of memory cells. The declination of induced neutralizing antibodies is ubiquitous in almost all virus infections, such as Zika [36], Dengue [29], and SARS-CoV-2 [37]. However, this declination is always discontinuous after their concentration drops to a stable level. This is contributed by the existence of memory cells [38]. Immunologists regarded the long-lasting B-cells or T-cells as “memory cells.” Experiments gradually demonstrated that although the so-called “memory cells” are some specific forms of immune cells [39,40], they have similar half-lives as normal CD8+ cells [41]. Therefore, maintaining antibody levels in “memory cells” should be explained as a state of equilibrium between decay and regeneration. We suppose this final equilibrium state derives from environmental antigen-like substances. Due to the presence of cross-interaction [42], the neutralizing antibody, no matter how specific it is, could have weak binding with other macromolecules in the solution. Those weak binding partners are defined as “environmental antigen-like substances” in our model; they could be food-resourced, air-resourced, or self-resourced. The presentation of those substances to T cells would give weak proliferation signals to neighboring cells. Therefore, a lower bound for specific neutralizing antibodies is present due to the positive feedback regulation of environmental antigenic substances. This hypothesis could explain why some antibodies persist in the body at detectable levels for a long time. We propose that the presence of environmental antigenic substances allows us to produce lifelong immunity to some pathogens.

The antibody level will eventually drop to zero based on the first model described in Equations (11)–(13). This implies that the antibody would finally fade in a short time. However, some antibodies can persist in the human body for a long time and provide lifelong protection, which is why we use vaccines. Following the last rationale, a second model is constructed to integrate environmental antigenic substances. The protein sequence of environmental antigen-like substances is ordinarily close to our own body and less antigenic, leading to a weak T-cell proliferation signal. An equilibrium state can be reached when the decay of memory cells is equal to their new synthesis. A good fit between the two scenarios further improves the reliability of this model. The first is that those environmental antigen-like molecules would help maintain the antibody at a certain level. This scenario can be easily recaptured in our second model. Our model can also explain the second scenario: environmental antigen-like substances cannot naturally elevate the antibody level to the final equilibrium concentration. Environmental antigenic substances can only help maintain the antibody level instead of triggering the increment. An exception is the allergy reaction, which happens under extreme conditions with the massive presence of environmental antigenic substances, especially when they have a strong binding affinity. Our second model can also recapture this phenomenon by increasing *p* and $k_{7}$ values.

Although the absolute equilibrium of specific antibodies is challenging to simulate mathematically, the presence of environmental antigen-like substances dramatically attenuates the antibody decay rate. The simulation results are shown in Figure 3. The decay rate of antibodies is closely related to the concentration of the antigen-like substance in the environment. The high concentration of the environmental antigenic substance would lead to a slow decay speed. The antibody decay rate is significantly slower (shown in the solid yellow curve) when there are massive environmental antigen-like substances, as shown in Figure 3A. The antibody level attenuation is also significantly influenced by the binding kinetics between environmental antigen and its corresponding antibody. The antibody decay rate is significantly slower (shown in the dashed yellow line) when $k_{3}$ sets a considerable value in Figure 3B. It represents a better binding kinetics between environmental antigen-like substance and its corresponding antibody.

It is common sense that environmental antigenic substances alone can hardly trigger antibody proliferation without pathogenic antigens. Numerical simulations also indicate that antigen-like substances in the environment hardly stimulate antibody proliferation. One scenario in which environmental antigen-like substances do not trigger antibody growth is recaptured in Figure 4A. The insufficient capacity of environmental antigens to induce antibody magnification derives from their poor binding affinity toward antibodies. In most cases, the presence of antigen-like substances in the environment will not directly stimulate antibody proliferation but will significantly attenuate the decay rate after antibody proliferation, as shown in Figure 4A. The situation can be altered when the antigen-like substances are sufficient or their binding affinities toward some antibodies are powerful. Antigen-like substances in the environment can also stimulate antibody levels, as shown in Figure 4B. This might help elucidate the underlying mechanism of allergy. A significant difference between environmental antigen-like substances and pathogenic antigens is that the former are deficient in self-replication. However, their concentrations are maintained at a relatively stable level due to constant replenishment from the environment.

### 3.2. Characteristics of Immune Response after Infected with Different Virus Strains

For the same type of virus, highly virulent strains have high replication activity [43]. Infection with highly virulent variants will lead to a prominent peak viral load and can exhibit strong transmissibility per time unit. However, its infection cycle is typically short. These infection dynamics are defined as acute infections in clinical performance [44]. The cytokine storm induced by a quickly elevated antibody level is responsible for the acute response [45]. Weak virulent strains have the opposite behaviors, which could lead to chronic infection in clinical settings. It does not typically induce a quick immune response, resulting in a more extended infection cycle. All those situations can be recaptured using our model, which is displayed in Figure 5.

It can be noticed from Figure 5 that, under the exact initial viral invasion dosage, the infection caused by a less virulent strain has a more extended latency phase. Its peak viral load is significantly lower than a more virulent strain. The maximal level of the yellow line is significantly lower than that of the dashed blue line, indicating that the antibody production induced by the weak virus infection is significantly lower than that induced by a virulent strain. It is also reflected in Figure 5 that the infection cycle caused by the weak virus is remarkably longer than its strong counterpart.

### 3.3. How the Immune System Screens for Highly Binding Antibodies

Our immune system can automatically detect suitable binding antibodies and selectively proliferate them. Simulating the antibody screening process by mathematical modeling remains challenging for computational biology researchers. This study provides a preliminary attempt to elucidate this process. Our model suggests a strong correlation between the proliferation potential of a specific antibody and its binding kinetics. Antibodies with fast binding rates would obtain rapid proliferation, while those with low binding rates proliferate slower. Thus, the antibodies with fast binding rates can grow faster and dominate in the final antibodies’ composition, although various antibodies are triggered to proliferate during infection. Generally, antibodies with fast binding capacity have a more robust binding affinity. However, this relationship is not absolute. Binding affinity is the ratio of forward binding to reverse dissociation [46]. Infected individuals would have antibodies with good binding kinetics after infection [47,48]. Those antibodies display individual differences in binding affinity, with some antibodies having relatively stronger binding affinity [49]. We propose that the dominant factor affecting the efficiency of antibodies is not their absolute binding affinity $K_{d}$ but the binding reaction constant $k_{2}$. Both $\;k_{2}$ and $k_{2}$ have physical meaning in Equation (12): $k_{2}$ stands for the forward reaction constant, while $k_{2}$ represents the reaction constant of the reverse one. The dissociation constant $K_{d}$ = $k_{2}$/$k_{-2}$ can be used to evaluate the concentration of each component in an equilibrium state. $K_{d}$ can also be transformed into the binding energy between those two macromolecules and thus is an important indicator that describes the binding affinity between two molecules [50].

As shown in Figure 6, the antibody with the fastest binding rate (solid blue line in this figure) demonstrates a maximal proliferation magnitude. This selection will eventually lead to a high proportion of fast-binding antibodies in the final antibody composition. Although these five antibodies have the same binding affinity of $K_{d}$ = 1 × 10^−9^, their proliferations vary because of the differences in their binding speeds, and the antibody with the faster binding rate will gradually dominate. This example illustrates that the driving force in antibody reproduction is its binding speed, not its binding affinity. A different $k_{2}$ value and $k_{-2}$ value is applied to keep the dissociation constant $K_{d}$ as a fixed number. It is subversive to claim that the immune system would like to select those fast-binding antibodies rather than tightly binding antibodies. Experimental researchers always seek an antibody with an excellent binding affinity [47,48,49]. However, our model indicates that a solid-binding antibody might not perform well in preventing reinfection. Its kinetic behavior is more critical than its static binding affinity.

### 3.4. High Concentrations of Weakly Binding Antibodies Can Provide Effective Protection

It has been statistically discovered that vaccination with other vaccines, such as the influenza vaccine, can also provide some degree of protection against SARS-CoV-2 [51,52,53,54]. The third model is utilized to explain this phenomenon. Those parameters used in Figure 7 are defined in Section 2.3.

Figure 7 illustrates that the presence of weakly binding antibodies can produce an inhibitory capacity against the virus. The concentration of the weakly binding antibody increases from one in case 2 to a thousand in case 1. The maximal value of the dashed yellow line is smaller than that of the solid blue line, indicating a more substantial inhibition effect on virus reproduction. It can be noticed that the maximal level of the solid-binding antibody also decreases with the enhanced initial level of the weakly binding antibody. A particular concentration of weakly binding antibodies can simultaneously inhibit the proliferation of strongly binding, efficient antibodies in the host.

This example suggests that vaccinated people will always have better protection against reinfection than non-vaccinated people, regardless of the mutation of the virus. However, when we aim to stimulate the production of high levels of firmly bound antibodies through vaccination, the presence of weakly bound antibodies might interfere with this process. The vaccination effect on each individual might vary due to the weak antibody inference. Some individuals can induce sufficient fast-binding antibodies, while others cannot, due to differences in their initial antibody library. The simulation results indicate antibody interferences might be greatly enhanced in people vaccinated against other viruses. This explains why some vaccines may occasionally lead to a higher overall mortality rate [14,55]. Keeping a relatively high antibody level for a specific pathogen, such as SARS-CoV-2, will shelter more people from infection. However, it might also have the side effect of decreasing the efficiency of other vaccines. The long-term effect of repeated booster vaccinations on SARS-CoV-2 needs a comprehensive study.

### 3.5. Calculation of the Protection Time Brought by Natural Infection

The prediction of protection time remains to be elucidated using a computational approach. The calculation of protection time is presented in this section with the application of a numerical approach. Figure 8 illustrates how we calculate the protection time upon natural infection. Details about how we calculate the protection time are described below:

First, we model the dynamic behaviors of all components (virus, virus–antibody complex, antibody). An initial number is assigned to each component before the first infection. For instance, *z*(0) = 1 stands for one invading virus. The virus number would vanish to zero soon after the antibody booms. The virus number dropped to zero after the 166th simulation time point, as illustrated in Figure 8.

After the antibody completely removes the virus, the second invasion is performed. The initial virus number would be manipulated to a non-zero integer at each following time unit, starting from 166 to 800. The early invasion would not lead to virus proliferation because the antibody level at that time was still high. However, the protection effect will gradually fade as the antibody level decreases. The earliest invasion that could lead to a virus proliferation is at the 326th time point in Figure 8. It is a time point when a breakthrough infection can happen. The time between the first infection and this time point is the protection duration brought by the first infection. Therefore, the protection time can be roughly estimated from 166th to 326th. We can also notice that the maximal viral load in the second infection is significantly lower than in the first infection. This indicates that the reinfection or breakthrough infection typically has a milder symptom than the first, although it will occur with a higher chance as time goes on. The protective effect will decrease as time increases, as shown in Figure 9 in the next section.

### 3.6. Factors Affecting the Duration of Antibody Protection: Concentration of the Environmental Antigen-like Substance, Viral Replication Capacity, and Antibody Binding Kinetics

Based on Equation (16), the virus will magnify itself when the $k_{-2}x-k_{2}yz-k_{5}z+k_{1}z,$ term is larger than zero. The analytic solution on the reinfection possibility is difficult to derive, although the antibody level and the virus replication capacity directly influence this term. A high concentration of environmental antigenic substances *p* and an excellent binding kinetic $k_{7}$ can help maintain the antibody at a relatively high level, thus providing an extended protection time. Meanwhile, the antibody kinetic features could also greatly influence the protection time, which is shown in Figure 9. A lifelong protective effect can happen when a virus has a weak replication capacity (small $k_{1}$ value), a high concentration of environmental antigen-like substances (large *p*-value), and a fast-binding antibody with an enormous $k_{2}$ value. Typical examples of this category are the smallpox virus [4], tetanus bacillus [56], and so on. In contrast, other viruses, such as influenza and SARS-CoV-2, may not be able to trigger a super-long protection time [5]. Meanwhile, for influenza and SARS-CoV-2 infections, there is a considerable variation in the protective time after natural infection or vaccination [57,58]. This variation is mainly attributed to the difference in their corresponding antibody kinetics.

As claimed in the previous section, the binding kinetic constant $k_{2}$ is much more important than the binding affinity $K_{d}$. Figure 9 illustrates how $k_{2}$ influences the protection time in different infection cases.

The second infection time is marked in Figure 9A, which starts around the 275th time unit. The protection time can be roughly estimated to be between 125th and 275th time unit. The protection time is significantly extended (100th to 1080th time unit) in Figure 9C when a better binding antibody is introduced. A virus loading threshold for the severe case is arbitrarily set as 6.4 × 10^5^ marked by the red arrow in Figure 9B. The protection time against severe infection can be derived based on Figure 9B,D, which is the time interval between those two green lines. Figure 9D demonstrates that the severe infection would happen after the 975th time unit with the same severe threshold. The protection time can be roughly estimated to range from 125th to 975th time unit. This protection phase is much longer compared to Figure 9B. This example demonstrates that an excellent binding antibody will protect people longer than the weak one. It indicates that the protection time brought by infection or vaccination could significantly vary for different individuals. The primary reason lies in the differences in the binding kinetics of antibodies–virus interaction. Another important discovery in this modeling is that people who generate suitable antibodies with fast binding kinetics rarely display significant symptoms in the first infection, which can be reflected in the smaller virus peak loading amount in the first infection in Figure 9C compared to Figure 9A. It is suggested that those fast-binding antibodies typically exist in patients without intense infection symptoms. Repeated vaccination might be limited in reshaping the composition of the final antibody reservoir, i.e., it does not significantly alter the binding activity of the antibodies. Therefore, targeted improvement of antibody binding activity against certain viruses is the key to extending vaccine protection in the long term. The heterogeneity of the individual antibody library might cause significant differences in the binding activity of antibodies to the same viral infection or vaccination, leading to a significant variation in protection duration. Therefore, we may need to change the vaccination strategy. Vaccination should induce the production of antibodies with high binding activity instead of a random incitation. The targeted induction of specific antibodies using gene editing provides a potential solution [59,60,61].

### 3.7. Parameter Estimation in Real Scenario

According to the data from clinical experiments, we can further fit the specific parameters to obtain the dynamic characteristics of antibodies within certain populations. The core content of this model is to accurately simulate the trend of mutual change between antibodies and viruses, so it is not required to add units to the mathematical model. By adjusting different parameters, especially the mean, and variance of the parameters, we can get the characteristics of the antibody dynamics behavior of a population. By comparing the available statistical data, we can estimate the distribution of antibody dynamics parameters of the whole population. Although these parameters have no direct physical meaning, they can more accurately reflect the decline of immunity in different individuals and the possibility of reinfection at different times.

We treat the concentration of the environmental antigenic substance as a constant value. The primary alternation is the difference in the binding capacity of the antibody in different individuals, which is defined as $k_{2}$ in our model. The binding kinetic between the environmental antigenic substance and the antibody, defined as $k_{7}$ in our model, varies among different people. The antibody kinetic performance characteristics of the population obtained using the above parameter combination are shown in Figure 10A. It was demonstrated that IgG antibody levels to the SARS-CoV-2 nucleocapsid waned within months, according to six months of data from a longitudinal seroprevalence study of 3217 UK healthcare workers [62]. The simulation result has a good match with this clinical report. The main factor determining the peak antibody concentration is $k_{2}$, while the driving force determining the antibody waning speed comes from $k_{7}$. The change in the overall antibody level of the population over time is shown in Figure 10A, and the change in the overall protective efficacy is shown in Figure 10B. According to this model, we can judge that in the absence of virus mutation, the overall protective efficacy of an initial 100% efficacy of the COVID-19 vaccine in a human population (10,000 human simulations) would drop to 97.21% after 100 days, 65.44% after 150 days, 39.28% after 200 days, and 28% after 240 days. The simulation results in Figure 8B are consistent with the clinical data on vaccine effectiveness. Based on a cohort study among US veterans, for the period 1 February 2021 to 1 October 2021, vaccine effectiveness against infection (VE-I) declined over time (*p* < 0.01 for time dependence), even after adjusting for age, sex, and comorbidity. VE-I declined for all vaccine types, with the most significant declines for Janssen, followed by Pfizer-BioNTech and Moderna. Specifically, in March, VE-I was 86.4% for Janssen, 89.2% for Moderna, and 86.9% for Pfizer-BioNTech. By September, VE-I had declined to 13.1% for Janssen, 58.0% for Moderna, and 43.3% for Pfizer-BioNTech [63]. According to a retrospective cohort study on the effectiveness of the mRNA BNT162b2 vaccine, effectiveness against infections declined from 88% (95% CI 86–89) during the first month after complete vaccination to 47% after five months. Among sequenced infections, vaccine effectiveness against infections of the delta variant was high during the first month after complete vaccination (93%) but declined to 53% after four months [64]. Similar trends were observed in the cohort study conducted in Qatar [65].

### 3.8. Recovered Patients with Retest Positive for SARS-CoV-2

In SARS-CoV-2 infections, we noticed some rare occurrences of reinfections by patients themselves [66]. This recurrent positivity is not caused by the invasion of environmental viruses but by the re-proliferation of internal viruses [67]. The self-reinfection may lead to a second epidemic outbreak, making it more challenging to prevent and control the epidemic. The scenario of self-reinfection is a frequent phenomenon in virus infection. Many people suffer from recurrent respiratory infections [68]. It is also typical when HBV or HCV infection occurs [69]. Our model suggests a possible mechanism underlying this phenomenon. The patient will experience reinfection under certain parameter sets, as shown in Figure 11.

This example above shows that it is difficult to completely eliminate a specific pathogen in the presence of fast-binding antibodies. However, the peak viral load during the infection is small, as shown in the solid red line in Figure 11A. This will lead to a relatively mild symptom. The peak viral load in Figure 11A is significantly lower than the typical infection cases illustrated in Figure 8 and Figure 9.

The low virus load does not have an intense stimulation on antibody production. Therefore, it is impossible to eliminate the virus from the body due to insufficient antibody quantity. The virus will regain the opportunity to proliferate when the antibody level decreases later. The patient may display a positive nucleic acid test result again. Multiple virus resurgences are marked in the dashed green cycle after the first infection, as displayed in Figure 11A. However, the infections in those cases are generally less symptomatic and even asymptomatic. A more extreme case is a long-term chronic infection, such as the appearance of a long-positive patient [70]. Like self-reinfection, an equilibrium state can be reached if the antibody–antigen interaction is moderate in those long-positive patients. In this case, pathogens would not be eliminated but maintained at a relatively stable level, forming a chronic infection. The low concentration of pathogenic antigens only provides a limited driving force for promoting antibody reproduction. The antibody and the virus will remain relatively low for a long time, as reflected in Figure 11B.

Pathogenic antigens and environmental antigenic substances all contribute to chronic inflammation. For chronic infections caused by pathogenic microorganisms, a short-term boost in antibody levels can be used to accomplish a complete clearance of pathogenic microorganisms [71]. Chronic infections could permanently disappear or significantly improve after they become acute infections in the clinic. The chronic symptom will disappear or become invisible after the healing from the acute infection [72]. Conversely, chronic inflammation caused by environmental factors can be removed by shielding environmental antigens for a certain period [73]. The blockade of antibody–antigen interaction will decrease the corresponding antibody level, thereby attenuating or eliminating the immune response. An allergic reaction can be significantly alleviated in the upcoming contact after this treatment.

## 4. Discussion

The application of mathematical modeling displays good epidemic forecast capacity at the population level [22,23,74,75]. Mathematical models are also helpful in quantitatively elucidating the immune process’s dynamics on an individual level. We are motivated to develop a new mathematical model that can explicitly model the antibody dynamics based on those pioneering modeling attempts [17,18,19,20,21,24,25,26,27,28,29,30]. A computational antibody dynamic model is finally proposed, which can help us explain some phenomena mentioned in the introduction.

I.How are memory cells maintained?

The environmental antigen-like substances maintain memory cells in our model. It was demonstrated that, in the absence of a pathogen, the environmental antigen could not directly stimulate antibody proliferation in most cases except in allergic reactions. The concentration of memory cells is closely correlated to the property and concentration of its corresponding environmental antigenic substances. Some memory cells against a specific pathogen can be maintained at a high level for decades. The persistent stimulation by the environmental antigenic substances, rather than the eternal lifespan, leads to the long-term existence of those immune memory cells. The existence of super-binding antibodies and high concentrations of environmental antigens can also trigger allergic reactions.

II.How does our immune system screen for antibodies with solid binding affinity?

Based on the model, antibodies with faster binding kinetics would increase proliferation. The binding constant is not linearly correlated to the binding affinity. The immune system tends to select antibodies with fast binding kinetics rather than solid binding affinity. The underlying mechanism comes from the positive feedback regulation of the virus–antibody complex. A good binding antibody would lead to a faster generation of the virus–antibody complex. This complex would further promote the proliferation of its corresponding antibody.

III.Why do people vaccinated by the influenza vaccine or other vaccines have a lower mortality rate from SARS-CoV-2 infection?

An interesting phenomenon during the COVID-19 pandemic is that the mortality rate is significantly lower in people who have received the influenza vaccine and other vaccines than in their unvaccinated counterparts [51,52,53,54]. The non-specific binding antibody level can be elevated after non-specific vaccination. Although weakly binding antibodies inhibit the proliferation of firmly binding antibodies in vivo, they can significantly inhibit virus proliferation and reduce peak viral load. This would lead to a lower mortality rate and milder symptoms. The potential concerns are also discussed in Section 3.4. A particular concentration of weakly binding antibodies can inhibit the proliferation of strongly binding antibodies, as revealed in Figure 7. The rationality of repeated booster vaccination needs a comprehensive evaluation because it might weaken the immune response to other pathogens.

IV.How could we effectively calculate the protection duration of a specific antibody?

Concerns are remarkably booming when people realize the protection of the SARS-CoV-2 vaccine has declined over time. Given the experimental data on viruses and antibodies, the kinetic parameters of this model can be inferred. The protection time of the vaccine or natural infection can be further deduced. Personalized prediction is also feasible with the availability of individual antibody behavior. The protection times of different individuals can be immensely varied, ranging from extraordinarily durable to very transient. Three factors that influence the protection time are discussed in Section 3.6. The virus’s attributes, such as the replication speed, influence the antibody production time. A faster-replicating virus, for instance, an RNA virus, inclines to have a shorter antibody protection time after natural infection. Virus immunogenicity against T-cells influences the duration of antibody protection. Viruses with strong T-cell immunogenicity provide a strong stimulus for antibody proliferation, exhibiting extended and even lifelong protection. The antibody binding kinetics also contribute to the difference in its protection performance, as illustrated in Section 3.6.

V.Why are there cases of self-reinfection?

Self-reinfection prevails in less virulent strains such as the Omicron variant. Our model provides a plausible mechanism underlying this unusual phenomenon. The individual might exhibit self-reinfection with the presence of a good binding antibody; repeated infection can also be displayed when a less virulent strain invades the patient. This can help explain why Omicron infections are inclined to be asymptomatic and self-reinfect.

An extreme case of self-reinfection is a chronic status in which the individual displays positive nucleic acid results for a long time. The chronic infection of SARS-CoV-2, together with other chronic infections, can also be explained by this model. It was demonstrated that a sub-equilibrium state could be maintained under specific parameter combinations with a relatively low antibody level, thus exhibiting long-term chronic inflammation. The long-term chronic infection is maintained by the constantly active (even weakly) antigens, which can come (i) from the non-coding human genome, where there are, for example, proteins of ancient viruses such as HER-V that produce antibodies a long time after their passage in the host genome, providing cross-protection against other infectious agents [76,77,78,79]; (ii) from the V(D)J mechanism of the innate immunity which can also give birth to antibodies with a large spectrum of actively participating to non-specific defenses against pathogens [80,81,82,83]; and (iii) from other vaccines (it is for example known that BCG reprograms the innate immunity [84]).

VI.Why do vaccinations show considerable differences in protection?

According to our antibody kinetic model, the antibody interference effect is the most important contributing factor to the differences in vaccine protection time, apart from viral mutations. Large amounts of interfering antibodies can inhibit the production of high concentrations of fast-binding antibodies after vaccination, causing a decrease in protection efficiency and protection period. The antibody reservoir’s heterogeneity in the human population also influences the generation of those fast-binding antibodies. The presence of good templates could promote the production of fast-binding mature antibodies after a few rounds of somatic hypermutation. Those fast-binding antibodies could provide extended protection. Environmental antigenic substances also influence the protection duration by influencing the antibodies’ decay rate. Clinical reports [85] have demonstrated that age and gender are statistically associated with differences in antibody response after vaccination. The IgG antibodies triggered by the SARS-CoV-2 BNT162b2 vaccine significantly varied among ages. Young people tend to generate more neutralizing antibodies compared to their elder counterparts. This mainly reflects a variation in $k_{4}$ values in our model. A faster antibody generation capacity, equal to a larger $k_{4}$ value, would confer a stronger immunity to younger people. However, it does not guarantee that the protection duration in young people would be longer than that in older people for each individual. As discussed in Section 3.6, the antibody attributes, together with environmental antigenic substances, strongly impact the protection time. Suitable antibodies with fast binding constant $k_{2}$ and high concentrations of environmental antigenic substances would prolong the protection time. The clinical report also indicates that females have stronger immunity than males. However, this trend is as insignificant as the influences of age. In this study, the forecast is performed to predict the population’s behavior, which does not consider the influences of age and gender. A more accurate and specific forecast can be performed in a future study when we integrate age and gender differences into the model. In this case, a different $k_{4}$ value would be assigned to each subgroup.

VII.How can we improve the protective efficiency and duration of vaccines?

Persistent exposure to massive antigenic substances can boost the neutralizing antibody level in a short period. However, the repeated booster can hardly magnify the proportion of solid-binding antibodies in the final antibody composition. The targeted induction of antibodies with excellent binding kinetics could provide prolonged protection against reinfection. Researchers are paving the way in this direction through gene-editing techniques [59,60,61]. Another concern with the constant booster shots is that the antibody interference might diminish the effectiveness of other vaccines, including the expected vaccines against massively mutated strains of SARS-CoV-2. Therefore, instead of pursuing a short-term antibody surge, we suggest that researchers aim to induce antibodies with fast-binding activity.

Besides providing a quantitative explanation of virus–host dynamics during the infection cycle, this model has several practical applications. One application is to predict the evolutionary direction of SARS-CoV-2 mathematically. The relationship between virulence and transmissibility can be simulated. A delicate equilibrium point that optimizes the transmissibility can be numerically obtained. Based on this model, we predict that the virulence of SARS-CoV-2 might further decrease, accompanied by an enhancement of transmissibility [86]. The second application is to optimize the vaccine inoculum dose mathematically. The antigen level, which can be represented as the initial inoculum dose, significantly influences the vaccine’s efficiency. A low dose might not be able to induce sufficient IgG to provide durable protection. In contrast, a high dose is more inclined to trigger a high level of IgG. However, it also has a more substantial adverse effect, representing a high antibody–virus complex level in this model. The high dose also increases its production cost. Therefore, a mathematical optimization can be performed to evaluate the optimal inoculum dose before clinical trials. Thirdly, this study could theoretically pave the way for future vaccine development. The protection duration of antibodies generated by natural infection is explicitly modeled in this study. The modeling of vaccines is different because the vaccine, independent of its type, does not replicate but is injected with an extremely high initial viral concentration. Different vaccines would have a different protection duration against breakthrough infection. vaccination efficiency is very complicated and generally weaker than the protection brought by a natural infection. The inactivation process for the inactivated vaccine would truncate its original structure, especially the epitope spots of spikes and nucleocapsid proteins. Its efficiency might be less than the mRNA vaccine. The vaccine based on the viral vector also generated deficient antigens compared to the real virus. After all, two major factors influence the protection duration of different vaccines: the peak level of neutralizing antibodies and viral mutations. A large dose of inoculum would trigger more neutralizing antibodies, leading to more extended protection and stronger adverse effects caused by the immune response. All vaccines would display weakened or even zero protection against variants with tremendous mutations due to a reduced $k_{2}$ value in this model. mRNA vaccine is better than the traditional vaccine because it can gradually generate realistic antigens. The traditional concept for vaccine development, with the application of an inactivated virus, viral vector, or mRNA, strictly forbids the use of a live virus. Our model demonstrated that the attenuated virus with proliferation constraints (smaller $k_{1}$ value in our model) could induce the immune response and antibody level in a milder way (shown in Figure 5). Those attenuated live viruses form a solid basis for asymptomatic infections [87]. There are at least two remarkable advantages of attenuated live viruses in future vaccine development. Firstly, since our model demonstrates that the peak antibody level is independent of the initial virus concentration, a minimal dose of live viruses could elevate specific IgG to a high level. It is also more infectious than traditional vaccines. This means a large number of populations can be covered with a few vaccination attempts. Secondly, despite the unlikely inactivation process, the attenuated virus is structurally identical to the original virus. Therefore, the specificity of neutralizing antibodies triggered by those attenuated viruses is the same as that of those induced by natural infections. As demonstrated by our model, an excellent binding capacity could induce a more durable protection time. The safety of those attenuated live viruses can also be guaranteed based on the principle of RNA replication: truncated mRNA cannot produce fixed offspring (full genome size).

Nevertheless, we also have to admit that our model has many hypotheses and uncertainties. Furthermore, it still lacks an in vivo experiment data fitting process, while many of the predictions and theories in the article remain to be confirmed experimentally and statistically in the future.

## Figures and Tables

**Figure 1 viruses-15-00586-f001:**
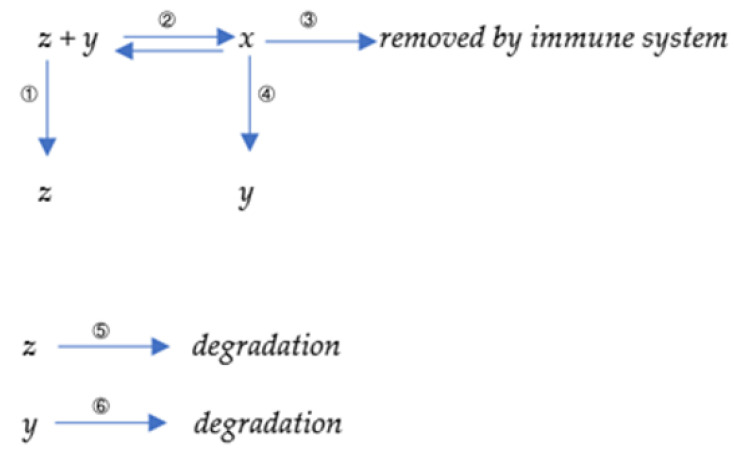
A simple diagram of host-virus interaction.

**Figure 2 viruses-15-00586-f002:**
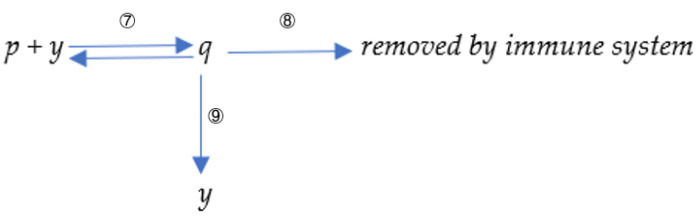
The role of environmental antigens in immune response.

**Figure 3 viruses-15-00586-f003:**
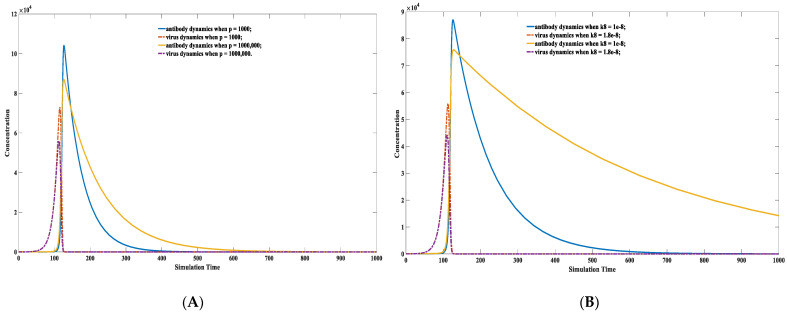
Antibody and virus dynamics modeling using different *p*(0) value. *p*(0) represents the concentration of environmental antigens. The virus-antibody dynamics are modeled under different environmental antigen concentrations (**A**) and different environmental antigen attributes (**B**). As shown in (**A**), the antibody decay rate is significantly slower (shown in solid yellow curve) when there is a large amount of environmental antigen-like substances. The parameter set we used is: *x*(0) = 0, *y*(0) = 100, *z*(0) = 10, *p*(0) = 1000 or *p*(0) = 1,000,000, $k_{1}=0.1,\;k_{2}$ = 1 × 10^−5^, $k_{-2}$ = 1 × 10^−14^, $k_{3}$ = 1, $k_{4}$ = 2, $k_{5}$ = 0.02, $k_{6}$ = 0.02, $k_{7}$ = 1 × 10^−8^, $k_{-7}$ = 1 × 10^−14^. As shown in (**B**), the antibody decay rate is significantly slower (shown in solid yellow curve) when $k_{8}$ sets a large value which corresponds to a better binding kinetics between environmental antigen-like stuff and its corresponding antibody. The parameter set we used is: *x*(0) = 0, *y*(0) = 100, *z*(0) = 10, *p*(0) = 1,000,000, $k_{1}=0.1,\;k_{2}$ = 1 × 10^−5^, $k_{-2}\;$= 1 × 10^−14^, $k_{3}$ = 1, $k_{4}$ = 2, $k_{5}$ = 0.02,
$k_{6}$ = 0.02, $k_{7}$ = 1 × 10^−8^ or $k_{7}$ = 1.8 × 10^−8^, $k_{-7}$ = 1 × 10^−14^.

**Figure 4 viruses-15-00586-f004:**
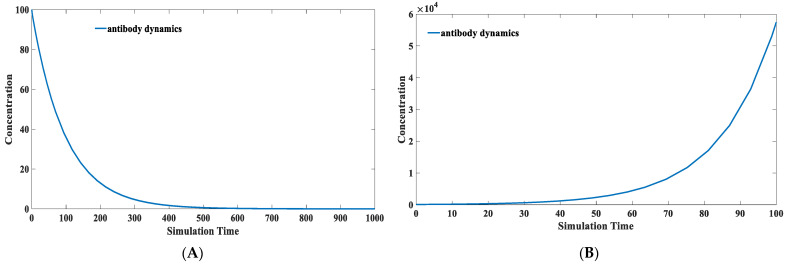
Antibody dynamics modeling with different $k_{7}\;$ values. (**A**) One scenario where environmental antigen-like substances do not trigger antibody growth. As shown in (**A**), the antibody does not engage proliferation due to the presence of environment antigen-like molecules. The parameter set we used is: *x*(0) = 0, *y*(0) = 100, *z*(0) = 0, *p*(0) = 1,000,000, $k_{1}=0.1,\;k_{2}$
*=* 1 × 10^−5^, $k_{-2}$*=* 1 × 10^−14^, $k_{3}$
*=* 1, $k_{4}$ = 2, $k_{5}$
*=* 0.02, $k_{6}$
*=* 0.02, $k_{7}$
*=* 1 × 10^−8^, $k_{-7}$
*=* 1 × 10^−14^. (**B**) One scenario where environmental antigen-like substances do trigger antibody proliferation. As shown in (**B**), the antibody does engage proliferation due to the presence of environment antigen-like molecules. The parameter set we used is: *x*(0) = 0, *y*(0) = 100, *z*(0) = 0, *p*(0) = 1,000,000, $k_{1}=0.1,\;k_{2}\;$= 1 × 10^−5^, $k_{-2}\;$= 1 × 10^−14^, $k_{3}$ = 1, $k_{4}$ = 2, $k_{5}$ = 0.02, $k_{6}$ = 0.02, $k_{7}$ = 1 × 10^−7^, $k_{-7}$ = 1 × 10^−14^. The antibodies might significantly increase when the environmental antigenic substances bind strongly with them. This always induces allergic reactions.

**Figure 5 viruses-15-00586-f005:**
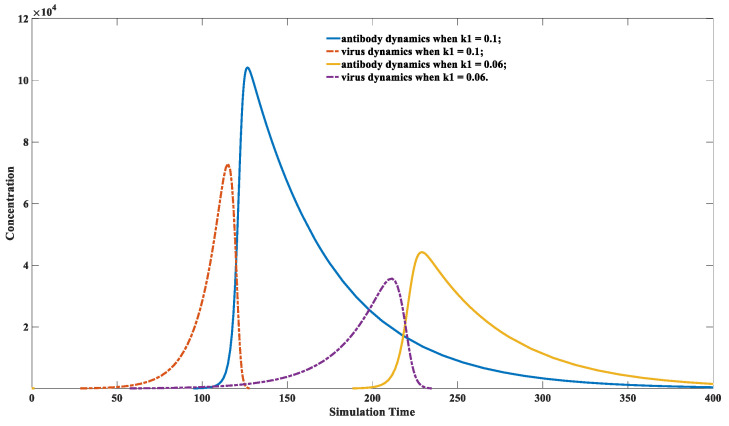
Different immune behaviors toward variants with different replication activities. The parameter set we used is: *x*(0) = 0, *y*(0) = 100, *z*(0) = 10, *p*(0) = 1000, $\;k_{1}=0.06\;or\;0.1,\;k_{2}$ = 1 × 10^−5^, $k_{-2}\;$ = 1 × 10^−14^, $k_{3}$ = 1, $k_{4}$ = 2, $k_{5}$ = 0.02, $k_{6}$ = 0.02, $k_{7}$ = 1 × 10^−8^, $k_{-7}$ = 1 × 10^−14^. As shown in this figure, the antibody response is milder when the host is infected by a less toxic strain (smaller $\;k_{1}$ value 0.06). The peak viral load is also less compared to its high toxic counterpart (bigger $\;k_{1}$ value 0.1). This indicates that even for the same virus infection, the immune response would vary greatly due to the differences in viral replication capacity of different variants.

**Figure 6 viruses-15-00586-f006:**
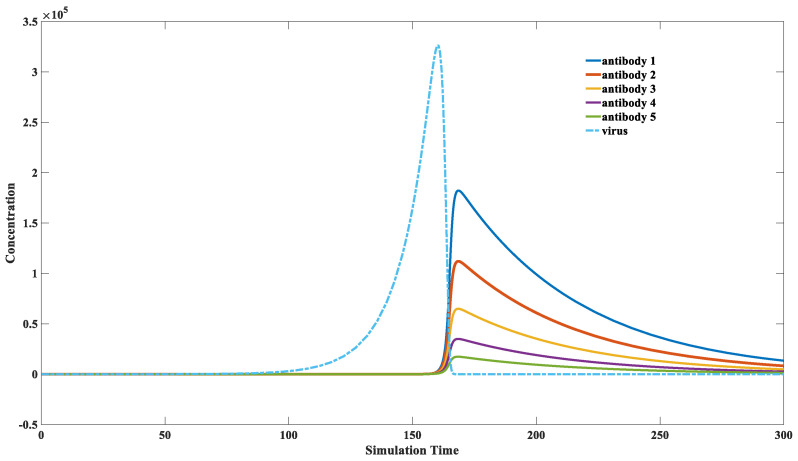
Dynamics of different antibodies with different kinetic attributes. The parameter sets we used are: *x*(0) = 0, *y*(0) = 1, *z*(0) = 1, *w* = 1, $k_{1}=0.1,\;k_{2}\;$ = 1 × 10^−5^, $k_{-2}\;$= 1 × 10^−14^, $k_{3}$ = 1, $k_{4}$ = 2, $k_{5}$ = 0.02, $k_{6}$ = 0.02, for antibody 1;$\;k_{2}$ = 9 × 10^−6^, $k_{-2}$ = 9 × 10^−15^ for antibody 2; $k_{2}$ = 8 × 10^−6^, $k_{-2}$= 8 × 10^−15^ for antibody 3; $k_{2}$ = 7 × 10^−6^, $k_{-2}$ = 7 × 10^−15^ for antibody 4; $k_{2}$ = 6 × 10^−6^, $k_{-2}$ = 6 × 10^−15^ for antibody 5. It is demonstrated in this figure that the faster-binding antibodies engage amplification at a greater magnitude. In this way, the immune system selects those specific neutralizing antibodies.

**Figure 7 viruses-15-00586-f007:**
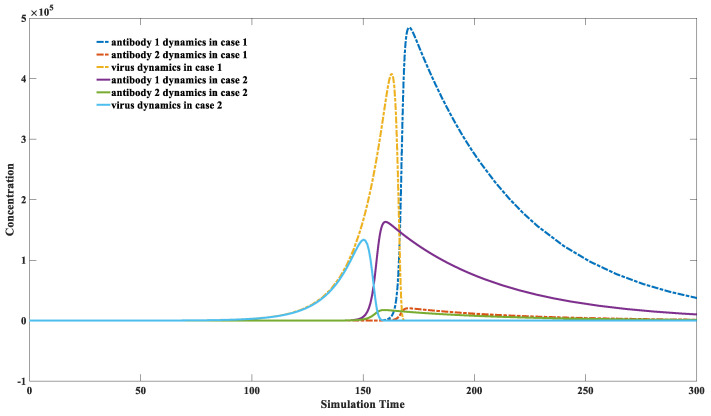
High concentrations of weakly binding antibodies can provide effective protection. A plot of the inhibitory capacity of a specific concentration of weakly binding antibodies against infection is presented. Two types of antibodies are presented in this figure: antibody 1 has a strong binding capacity ($K1_{\mathrm{on}}\;$
*=* 1 × 10^−5^) while antibody 2 has a relatively weak binding capacity ($K2_{\mathrm{on}}\;$ = 5 × 10^−6^). Two scenarios are simulated: both antibodies have low initial concentrations in case 1; weakly binding antibody has a high initial level in case 2. The parameter sets we used are: $x_{1}$(0) = $x_{2}$(0) = 0, $y_{1}$(0) = $y_{2}$(0) = 1, *z*(0) = 1, $w_{1}$ = $w_{2}$ = 1, $K1_{\mathrm{on}}\;$
*=* 1 × 10^−5^, $K1_{\mathrm{off}}\;$*=* 1 × 10^−14^, $K2_{\mathrm{on}}\;$
*=* 5 × 10^−6^, $K2_{\mathrm{off}}\;$
*=* 1 × 10^−14^, $k_{3}$ = 1, $k_{4}$ = 2, $k_{5}$ = $k_{6}$ = 0.02, $k_{1}$ = 0.1 for case 1; $y_{2}$(0) = 100 for case 2. It can be seen that the peak viral load and antibody level are significantly lower in case 2, which corresponds to a milder immune response. It indicates that the elevated weakly binding antibodies could also provide protection against severe infection.

**Figure 8 viruses-15-00586-f008:**
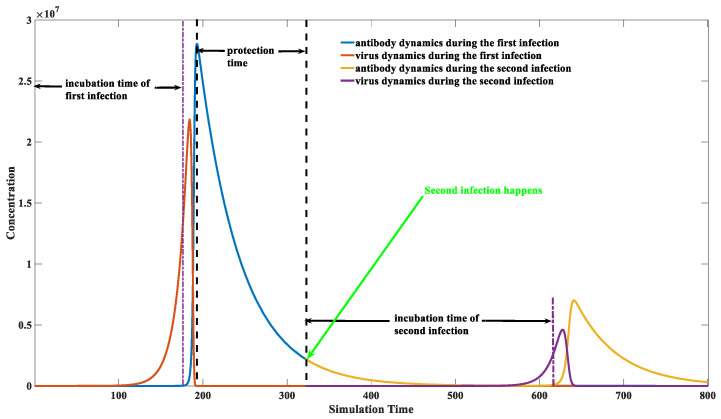
An illustration of protection time calculation. The parameter set we used is: *x*(0) = 0, *y*(0) = 100, *z*(0) = 10, *p*(0) = 10,000, $k_{1}=0.1,\;k_{2}$ = 1 × 10^−7^, $k_{-2}$ = 1 × 10^−14^, $k_{3}$ = 1, $k_{4}$ = 2, $k_{5}$ = 0.02, $k_{6}$ = 0.02, $k_{7}$ = 1 × 10^−8^, $k_{-7}$ = 1 × 10^−14^. Incubation time is calculated as the time interval between virus entrance and the production of antibodies. The viruses would engage a proliferation earlier than the antibodies. The patient is still asymptomatic even though the virus has reached a high level. Symptoms such as fever would appear when the antibody–virus complexes reach beyond a specific threshold. The second infection is marked with a green arrow. It can be seen in this figure that a second infection could occur when the IgG level drops below a certain level. It does not necessarily require a zero IgG level when a breakthrough infection happens.

**Figure 9 viruses-15-00586-f009:**
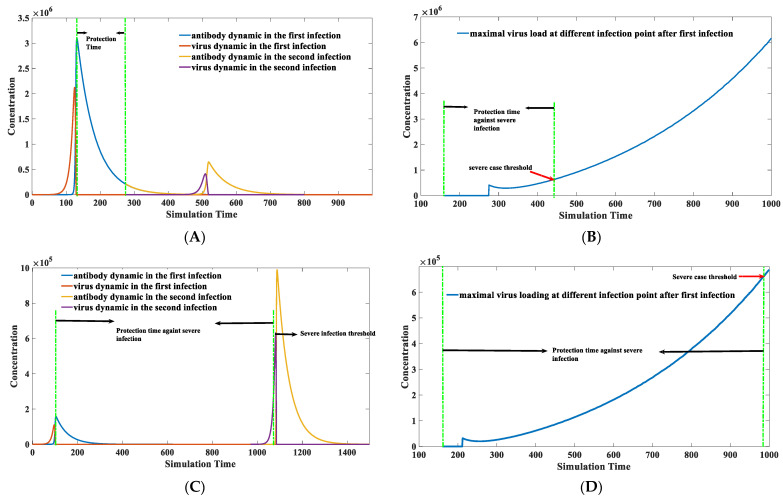
Protection time calculation when the antibody has a specific binding kinetic constant $k_{2}$. It can be inferred from this figure that a fast-binding neutralizing antibody would provide a longer protection time when we compared (**A**) with (**C**). The protection time against severe infection can also be prolonged, given the faster binding kinetics when comparing (**B**) to (**D**). (**A**) Protection time calculation when the antibody has a weak binding kinetic constant $k_{2}$ = 1 × 10^−6^. The parameter set we used is: *x*(0) = 0, *y*(0) = 100, *z*(0) = 10, *q*(0) = 1 × 10^6^, $k_{1}=0.1,\;k_{2}$ = 1 × 10^−6^, $k_{-2}$ = 1 × 10^−14^, $k_{3}$ = 1, $k_{4}$ = 2, $k_{5}$ = 0.02, $k_{6}$ = 0.02, $k_{7}$ = 1 × 10^−9^*,*
$k_{-7}$ = 1 × 10^−14^. (**B**) Maximal virus load at different infection points when the antibody has a binding kinetic constant $k_{1}$ = 1 × 10^−6^. The parameter set used is the same as (**A**). (**C**) Protection time calculation when the antibody has a strong binding kinetic constant $k_{2}$ = 1 × 10^−5^. The parameter set we used is: *x*(0) = 0, *y*(0) = 100, *z*(0) = 10, *q*(0) = 1 × 10^6^, $k_{1}=0.1,\;k_{2}$ = 1 × 10^−5^, $k_{-2}\;$= 1 × 10^−14^, $k_{3}$ = 1, $k_{4}$ = 2, $k_{5}$ = 0.02, $k_{6}$ = 0.02, $k_{7}$ = 1 × 10^−9^, $k_{-7}$ = 1 × 10^−14^. (**D**) Maximal virus load at different infection points when the antibody has a binding kinetic constant $k_{2}$ = 1 × 10^−5^. The parameter set used is the same as (**C**).

**Figure 10 viruses-15-00586-f010:**
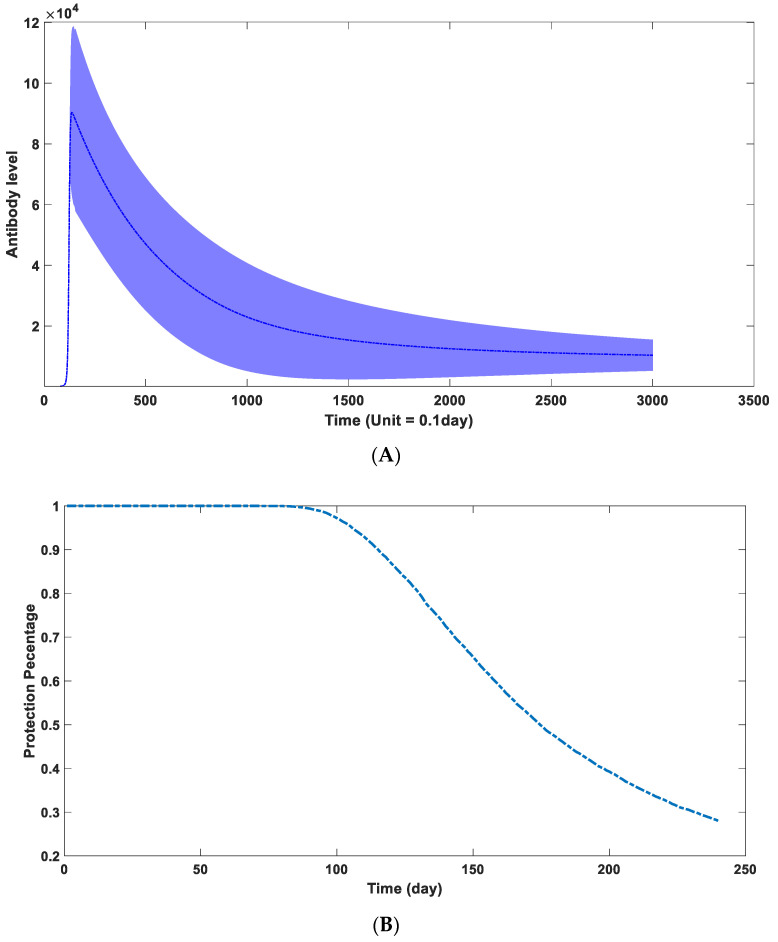
(**A**) the dynamic behavior of antibodies in the overall population through time. (The blue zone around mean curve stands for 95% confidence interval). It can be seen that the IgG level would significantly decline after reaching a peak level. However, its degradation does not follow a simple mathematical formula. Its descent rate would gradually decline and be maintained at a relatively stable level after 200 days. (**B**) The protection performance of antibodies in the overall population through time. It can be seen in this figure that the protection efficiency of induced neutralizing antibodies would be maintained at a relatively high level in the first 100 days. Its protection efficiency would engage a rapid decline after the first three months.

**Figure 11 viruses-15-00586-f011:**
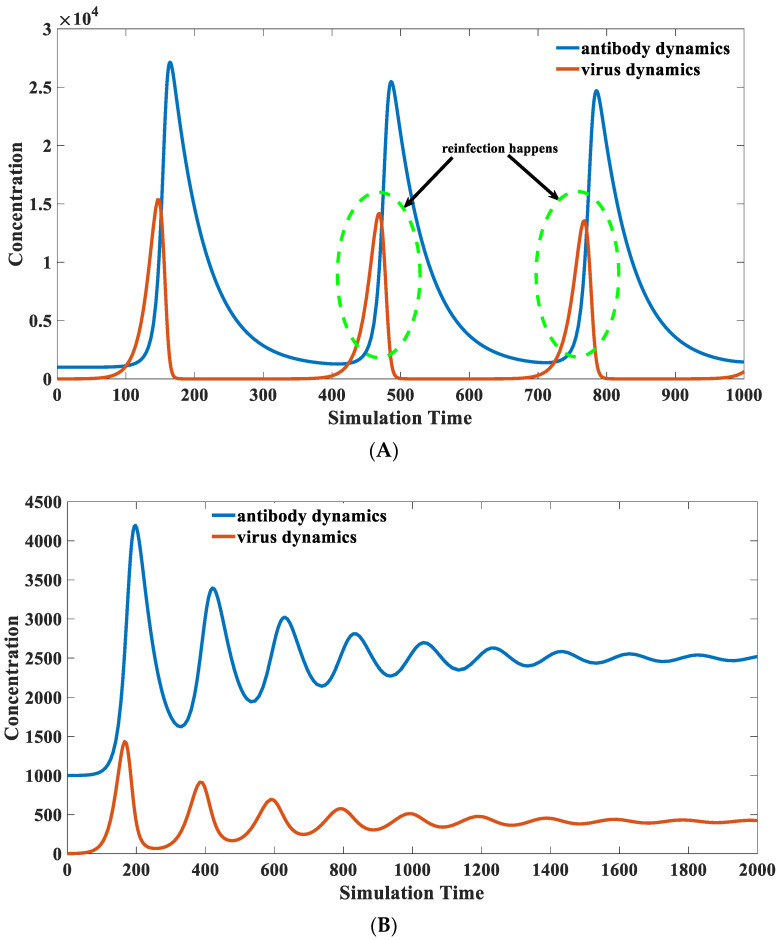
(**A**) Self-reinfection scenario. The parameter sets we used are: *x*(0) = 0, *y*(0) = 1000, *z*(0) = 1, *w* = 1000, $k_{1}=0.1,\;k_{2}\;$ = 1 × 10^−5^, $k_{-2}\;$ = 1 × 10^−14^, $k_{3}$ = 1, $k_{4}$ = 2, $k_{5}$ = 0.02, $k_{6}$ = 0.02. Reinfections are represented as repeated waves in this figure. It indicates that the viruses could re-proliferate when the antibodies cannot completely eliminate them. The viruses start to proliferate when the antibodies drop to a certain level, leading to self-reinfection in this case. (**B**) Scenario of chronic infection. The parameter sets we used are: *x*(0) = 0, *y*(0) = 1000, *z*(0) = 1, *w* = 1000, $k_{1}$ = 0.1, $k_{2}$ = 3 × 10^−5^, $k_{-2}\;$ = 1 × 10^−5^, $k_{3}$ = 1, $k_{4}$ = 2, $k_{5}$ = 0.02, $k_{6}$ = 0.02. In this case, pathogens would not be eliminated but maintained at a relatively stable level, forming a chronic infection. The low concentration of pathogenic antigens only provides a limited driving force for promoting antibody reproduction.

## Data Availability

Matlab source codes are available at https://github.com/zhaobinxu23/antibody_dynamics, accessed on 1 January 2022.

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
