# Peer review of "A Novel Mathematical Model That Predicts the Protection Time of SARS-CoV-2 Antibodies"

_viruses, 2023, doi:10.3390/v15020586_

Round 1

Reviewer 1 Report

Abstract is unclear and non-concrete.

No evidence of the practical application.

Pure graphical materials signs and explanations.

Reviewer 2 Report

Major concerns.

1. Is the model fit for all ages?
In children, susceptible cells may be lower than in adults due to body mass and the presence of ACE receptors. Moreover, the immunity in children is stronger than in adults and the elderly when using the same dose of vaccine.
In contrast to elderly that low immunity than other age groups.

If it was possible. Suggest discussing the age-associated immunogenicity and suitable age range for your model to make it more clear. 

2. What is the effect of sex?
In females, the immunity seems higher than in males.

Was the forecast trend with 95% CI coverage for both males and females?

3. What is the effect of different types of vaccines (inactivated, viral vector, or mRNA) on the forecast trend?

Different types could give different B- and T- cells profiles. Are the coverage of the parameter to all types of vaccines?

4. What is the effect of vaccine dose(s) on the antibodies profiles (trend, kinetics, and protection)?

Is the model answer the effect of dose, including booster dose and "anamnestic response"?

Minor concerns.

1. Line 30-31, please specify the specification date of summarised total confirmed cases and lethal because the total number of COVID-19 cases will increase daily.

Reviewer 3 Report

Dear Authors,

Your article entitled “A Novel Mathematical Model that Predicts the Protection Time of SARS-CoV-2 Antibodies” is a good one to contribute to the literature. I have read the full article with great interest. I would like to thank the authors and their collaborators for the time and effort involved in this work. This is an interesting article focusing on the mathematical model for protection time of SARS-CoV-2 Antibodies.  In my opinion, this is a straightforward report including references to the recent literature with regard to the virulence of SARS-CoV-2 and the antibody protection time. Moreover, this work presents sound implication for health policies and future research. I recommend its publication.

Author Response

Thanks a lot for your comments!

Round 2

Reviewer 1 Report

Abstract is still looks unclear.

Source of data explained unclear (the virological part has to be clear described - patients, numbers, Ab levels, methods and sites of lab testing)

Units of measurement in the graphical  presentation are non-concrete